# Waveguide Structure Design and Simulation for High-Temperature Corrosion Thickness Detection

Yan Li [1,2,*], Xi Luo [1], Ruihao Liu [1], Ze Yun [1] and Jixiang Zhang [1,2]

1　College of Mechanical and Transportation Engineering, China University of Petroleum Beijing, Beijing 100249, China
2　Beijing Key Laboratory of Process Fluid Filtration and Separation, China University of Petroleum Beijing, Beijing 102249, China
*　Correspondence: heartonelee@cup.edu.cn

**Featured Application: The paper designed an optimized waveguide structure for ultrasonic testing even at a high temperature 500 deg C. With the waveguide structure, an ultrasonic-guided wave testing device was developed. Measurement error increases with temperature if the wave velocity is regarded as a constant. A temperature-dependent method was applied to achieve high precision detection at high temperatures.**

**Abstract:** Equipment corrosion often happens in the petrochemical industry, especially when high temperature materials are transported. The corrosion phenomenon should be monitored as a leak may occur due to corrosion and even cause fires and explosions. However, ordinary ultrasonic testing is not suitable for high temperature conditions because the probe may break. A waveguide structure was designed to economically detect corrosion thickness even at a high temperature 500 deg C and avoid the failure of the ultrasonic probe. Based on the heat transfer simulation, a waveguide rod was determined with optimized material, length, width and thickness, and the experiment validated the calculated result. Then, ultrasonic propagation through the designed waveguide rod and specimen was simulated. Propagation, reflection, attenuation and dissipation of the ultrasonic wave within the combined structure were displayed. A clear ultrasound signal was found near the center, while signal attenuation and dispersion occurred as it is gradually far away from the center. With the waveguide structure, an ultrasonic-guided wave testing device was developed to measure the thickness at high temperatures. Measurement error increases with temperature if the wave velocity is regarded as a constant. A temperature-dependent method was applied to achieve high precision detection at high temperatures. The research has good application potential for the corrosion detection of high-temperature equipment.

**Keywords:** waveguide rod; high-temperature detection; heat transfer simulation; ultrasonic-guided wave testing; thickness measurement

## 1. Introduction

Oil and gas pipelines have a pivotal role in national economic development. However, corrosion issues can effectively diminish wall thickness, resulting in pipeline leakage and even explosion accidents. To guarantee the smooth and secure operation of the equipment, engineers pay close attention to the corrosion of hardware and the reduction in pipe thickness. Frequent detection is necessary to avoid security incidents.

Therefore, pipeline corrosion detection has resulted in being imperative. The traditional method is to determine the specific corrosion situation of the pipeline by measuring the thickness of the pipe wall and analyzing the relevant information. In recent years, magnetic flux leakage detection, eddy current detection and ultrasonic-guided wave detection technologies have been widely used in pipeline corrosion detection at home and

abroad [1,2]. Magnetic flux leakage (MFL) detection is less costly and can detect the whole circumference of the pipe, but it is prone to untrue signals amid the detection process and has low sensitivity [3–5]. Eddy current detection can be sensitive to automatic detection of pipeline defects, but the actual depth of detection conflicts with its sensitivity [2,6,7]. Ultrasonic-guided waves have the advantage of single-point excitation, long propagation distance and low attenuation. In addition, they are capable of accurately locating the corrosion site and covering a large area. Therefore, ultrasonic-guided wave testing (UGWT) is widely used for corrosion detection of large diameter thick-walled oil and gas pipelines [8–10]. UGWT can also be combined with the concept of structural health monitoring. Fromme et al. [11] developed a guided ultrasonic wave array to monitor structural integrity. Lugovtsova et al. [12] analyzed the guided wave propagation in a multi-layered structure. Diamanti et al. [13] applied UGWT to monitor the structural health of aircraft composite structures. Gao [14] conducted an in-depth study of ultrasonic-guided wave mechanics in composite material.

Ultrasonic-guided wave testing can be directly applied to oil and gas pipeline corrosion detection at room temperature. Studies of nonlinear guided waves used to detect microcracks in structures and to characterize materials have been reported [15,16]. However, because of the poor high-temperature resistance of ordinary piezoelectric ultrasonic probes, which can only operate normally below half of their Curie temperature, they cannot be directly in contact with high-temperature pipelines [17]. To solve this problem, several scholars [17,18] devoted themselves to studying high-temperature piezoelectric materials, such as bismuth titanate and lithium niobate materials, which can reach a Curie temperature of more than 500 deg C. The weakness of the present piezoelectric ultrasonic probe-based UGWT is that it cannot be directly applied to the thickness measurement of high-temperature pipelines. Therefore, scholars have turned their attention to making the temperature at the site of contact with the ultrasound probe cooler. Cross was the first to propose the use of waveguide rods as a temperature buffer structure for high-temperature specimens, applied for a patent, and then used incomplete welding. He used a cylindrical waveguide rod welded to the surface of the workpiece. Billson [19] proposed a non-contact method to measure the wall thickness of hot steel pipes. Li et al. [20] also used a waveguide rod of the same material as the pipe under test as a temperature buffer structure. She welded it vertically to the outer surface of the pipe under test in a fully welded manner. Cegla [21] designed a rectangular waveguide rod to isolate the sensor and piezoelectric device from the high-temperature measurement area without the need for additional cooling equipment. Gao et al. [22] and Wang et al. [23] both proposed a method to excite shear horizontal (SH) waves.

The waveguide rod is not only a structure for buffering high temperature but also a medium for guiding the wave transmission. In terms of buffering temperature, Gao [24] argued that the buffer structure should have a high melting point because it is in direct contact with the high-temperature pipe. Moreover, the heat transfer coefficient of the buffer structure is as small as possible to achieve an excellent cooling effect. In terms of the perspective of wave propagation, Gong et al. [25] believed that the buffer structure should be flexible. The waveguide rod itself does not generate vibrations to avoid traveling excess ultrasound signals. In addition to that, the waveguide rod can transmit the stress wave, and the waveform distortion is as tiny as possible so that the signal can be read easily. In terms of numerical simulation, some scholars were dedicated to studying the propagation process of ultrasonic-guided waves in plates and pipes [26,27]. Li et al. [26] conducted numerical simulations of the propagation characteristics of AE signals in waveguide rods using ANSYS and found that AE signals are affected by the shape and size of waveguide rods to varying degrees. The finite element method, as a numerical analysis method with a theoretical basis and widely used effects, can be used to solve problems in the field of acoustics that cannot be solved by analytical methods [28].

In this paper, a rectangular waveguide structure for buffering high temperatures was designed. Optimum material and size were determined to ensure the ultrasonic testing

at a high temperature of 500 deg C. Ultrasonic propagation through the designed waveguide rod and specimen was calculated with COMSOL Multiphysics. Wave propagation, reflection, attenuation, and dissipation in the combined structure were displayed. Based on the waveguide structure, an ultrasonic-guided wave testing device was developed to detect the thickness of two specimens at high temperatures. The measured thickness varied increasingly with temperature if the wave velocity was set as a constant. Therefore, a temperature-dependent method was successfully applied to achieve high-precision detection at high temperatures. The research shows good potential in the petrochemical industry to detect thickness variation due to high-temperature corrosion.

## 2. Computational Methods

Figure 1 shows the computational model, including a high-temperature flat specimen at the bottom and a rectangular waveguide rod at the top. The simulation process was based on the ultrasonic pulse reflection principle to measure the thickness of the specimen. The high-temperature specimen was fixed to the waveguide rod. The ultrasonic probe located at the top of the waveguide rod emitted a pulse signal, here using an SH-guided wave. The ultrasonic pulse signal propagated along the waveguide rod and the specimen to be measured, reflecting when it reached the bottom surface and generating a pulse echo. The pulse echo was also received by the same ultrasonic probe, and the specific waveform can be displayed by an oscilloscope or a curve recorder [29].

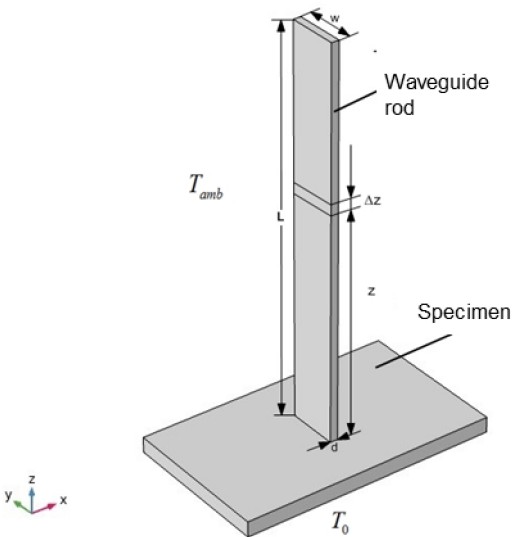

**Figure 1.** Computational model of the waveguide structure.

The thickness of the tested specimen can be determined by measuring the time of ultrasonic wave propagation in the material, which can be expressed as:

$$\delta = \frac{1}{2}c\Delta t - L \tag{1}$$

where $c$ is the propagation speed of the ultrasonic wave in the specimen; $\Delta t$ is the time difference between two adjacent wave peaks of the echo pulse signal; $L$ is the length of the waveguide rod.

The waveguide rod exists to avoid permanent failure of the ultrasonic probe due to direct contact with the high-temperature specimen and was in a state of steady heat conduction. The governing equation can be expressed as:

$$\nabla^2 t = 0 \tag{2}$$

In the model, the temperature of the specimen was determined. The bottom end of the waveguide rod was in contact with the high-temperature specimen, and heat transfer occurred with the air on the remaining surfaces. The ambient temperature was $T_{amb}$, the length of the waveguide rod was $L$, the width was $w$, and the thickness was $d$.

The flat specimen was stainless steel 316 L with a size of 250 mm $\times$ 180 mm $\times$ 50 mm, while the material, length and thickness of the waveguide rod needed to be designed by using COMSOL Multiphysics. Three materials were chosen as candidates, which are shown in Table 1.

**Table 1.** Three material properties used in the model.

| Parameters | SS316L | Aluminum | Copper |
|---|---|---|---|
| Density (kg/m$^3$) | 7800 | 2700 | 8960 |
| Young's modulus (GPa) | 211 | 70 | 110 |
| Poisson's ratio | 0.286 | 0.33 | 0.35 |
| Constant pressure heat capacity (J/(kg·K)) | 475 | 900 | 385 |
| Thermal conductivity (W/(m·K)) | 44.5 | 238 | 400 |

Figure 2 shows the schematic diagram of the mesh division. The waveguide rod used a structured mesh, and the specimen used a free tetrahedral mesh structure. The whole model contained 13,058 domain cells, 15,390 boundary elements and 920 edge cells. Grid sensitivity tests were validated. The ambient temperature was 20deg C and the temperature at the bottom surface of the specimen was 500 deg C. Define all the other surfaces of the specimen and the waveguide rod as the heat flow boundary conditions. At high temperatures, radiation also plays an important role, and it can be combined with the convection and take the empirical value of heat transfer coefficient as $h_{com} = 5 \, \text{W}/(\text{m}^2\cdot\text{K})$. Therefore, the thermal boundary conditions can be expressed as:

$$q = h_{com}(T - T_{amb}) \tag{3}$$

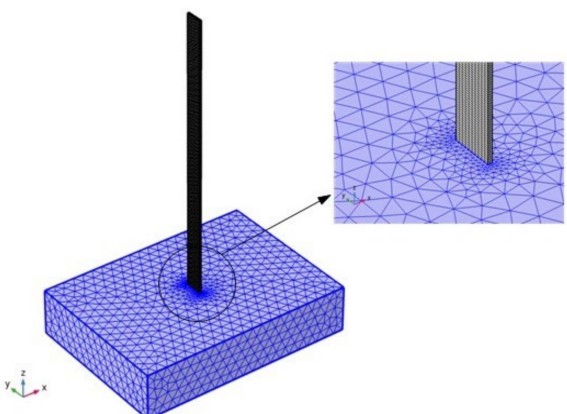

**Figure 2.** Mesh system of the model.

When the waveguide structure was determined, the same model was used to analyze the propagation of ultrasonic waves in the waveguide rod and the specimen. As the waveguide structure was fixed on the specimen by a clamping device with a strong pressing force, so the contact surface was coupled for the two parts in the model. The ultrasonic wave propagation was simulated using a time-domain display solver based on the interrupted Galliakin method (dG-FEM). The model used the velocity-strain formulation to solve the governing equations for a general linear elastic material [23,24] as:

$$\rho \frac{\partial \mathbf{v}}{\partial t} - \nabla \cdot \mathbf{S} = \mathbf{F}_{\text{v}} \tag{4}$$

$$\frac{\partial \mathbf{E}}{\partial t} - \frac{1}{2}\left[\nabla \mathbf{v} - (\nabla \mathbf{v})^{\mathrm{T}}\right] = 0 \tag{5}$$

$$\mathbf{S} = \mathbf{C} : \mathbf{E} \tag{6}$$

where v is velocity; $\rho$ is density; **S** is stress tensor; **E** is strain tensor; **C** is elastic tensor (or stiffness tensor); $\mathbf{F_v}$ is volumetric force.

Ultrasound is a mechanical wave. The propagation of ultrasound is the vibration of the medium that produces fluctuations. The propagation of sound waves is the propagation of energy. Hence, the process of ultrasound propagation follows Newton's second law, the conservation of energy and the law of conservation of momentum. Integrating the equation of state and the small amplitude approximation, we can represent the sound field in a solid elastic medium by the following linear equation:

$$\rho \nabla^2 u_i = C_{ijkl} u_{k,jl} + f_i \tag{7}$$

where $\rho$ is media density; $u$ is mass displacement; $C$ is elasticity factor; $f$ is external force; $\nabla^2$ is the Laplace operator.

Equation (7) is the active acoustic wave equation. If there is no external force, the equation can be expressed as:

$$\rho \nabla^2 u_i = C_{ijkl} u_{k,jl} \tag{8}$$

Equation (8) is also called the Helmholtz equation and is mainly used to calculate the propagation of sound waves in a semi-infinite space.

The equation for the sound field of an isotropic solid medium can be expressed as:

$$(\lambda + 2\mu)\nabla u - \mu \nabla^2 u = \rho \frac{\partial^2 u}{\partial t^2} \tag{9}$$

where $\lambda$ is Lame constant; $\mu$ is Lame constant.

Next, consider how to apply the excitation signal. In fact, the pulse signal was emitted by the piezoelectric ultrasonic probe and acted on the waveguide rod, so the Gaussian pulse signal was loaded to the upper end of the rod in the form of a load. Figure 3 shows the Gaussian pulse excitation signal with the functional expression in the following equation:

$$y(x) = A \times \cos(2\pi f_0 t) \times exp\left[-2\pi^2 (t-1)^2\right] \tag{10}$$

where $A$ is the signal amplitude; $f_0$ is the center frequency of the ultrasonic pulse signal, which is taken as 2.25 MHz.

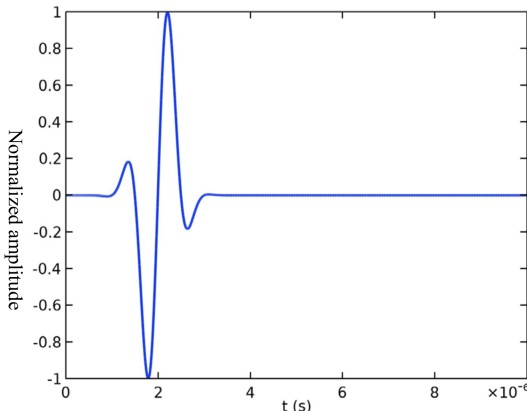

**Figure 3.** Excitation signal used in the model.

### 3. Results and Discussion

*3.1. Temperature Field*

3.1.1. Temperature Field of Waveguide Rods with Different Materials

Figure 4 shows the temperature field of the waveguide rod and the high-temperature specimen. From left to right, the waveguide rods are made of SS316L, aluminum, and copper, with the same dimensions. The materials and dimensions of the high-temperature specimens in the three models are also the same. Thermal analysis shows that the temperatures at the top waveguide rod are 33.6 deg C, 168.9 deg C, and 239.6 deg C, respectively.

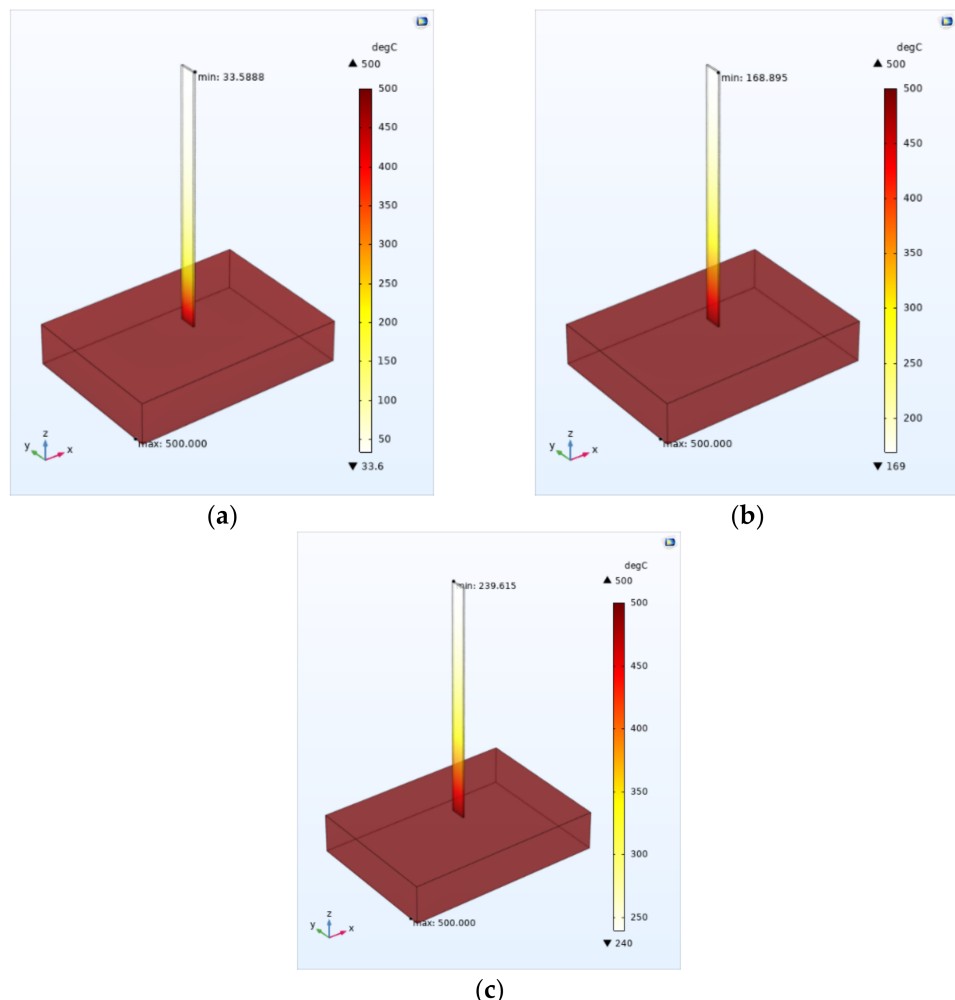

**Figure 4.** Temperature clouds of variable material models. (**a**) Temperature cloud of specimen and waveguide rod made of SS316L; (**b**) Temperature cloud of specimen and waveguide rod made of aluminum; (**c**) Temperature cloud of specimen and waveguide rod made of copper.

Figure 5 shows the temperature profile on the centerline of the waveguide rod. In order to compare the cooling effect of the three materials of the waveguide rod more visually, their curves are placed in the same graph. The left end of the temperature curve represents the highest temperature on the waveguide rod. The rightmost end of the temperature curve indicates the temperature at the top of the waveguide rod, which also is the lowest temperature. The starting temperature of the three curves is almost the same. In contrast, the temperature at the tip varies greatly, which is mainly determined by the material's physical properties. Comparing the differences of the cooling curves, we can find that the cooling effect of the waveguide rod made of SS316L is the best. When the length of the waveguide rod is 300 mm, it can meet the requirements of the experiment and ensure that

the ultrasonic probe is not affected by high temperature. Therefore, the waveguide rod for the experiment was made of SS316L and its length was 300 mm.

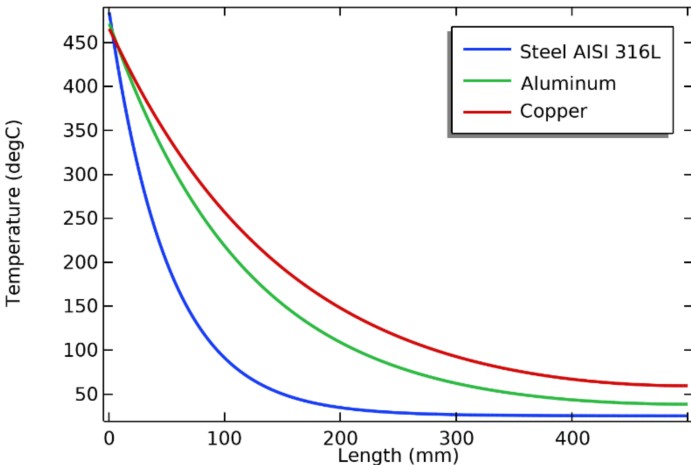

**Figure 5.** Temperature curves for SS316L, aluminum, and copper waveguide rods.

### 3.1.2. Temperature Field of Waveguide Rods with Different Sizes

Next, this article will further investigate the effect of the specific dimensions of the waveguide rod made of SS316L on its cooling effect. Carry out parameter scanning on five sets of width and thickness values, using the controlled variable method. The results of the parametric scans are shown in Figure 6. Figure 6a shows the temperature profiles of the waveguide rod at different width values, and Figure 6b shows the temperature profiles of the waveguide rod at different thickness values. The results show that the width of the waveguide rod is not the main factor affecting its cooling effect, while the thickness significantly affects its temperature buffering effect. Specifically, when the thickness of the waveguide rod reduces, the temperature drops faster. The laboratory used a waveguide rod with a width of 20 mm and a thickness of 1 mm, after considering the difficulty of factory processing.

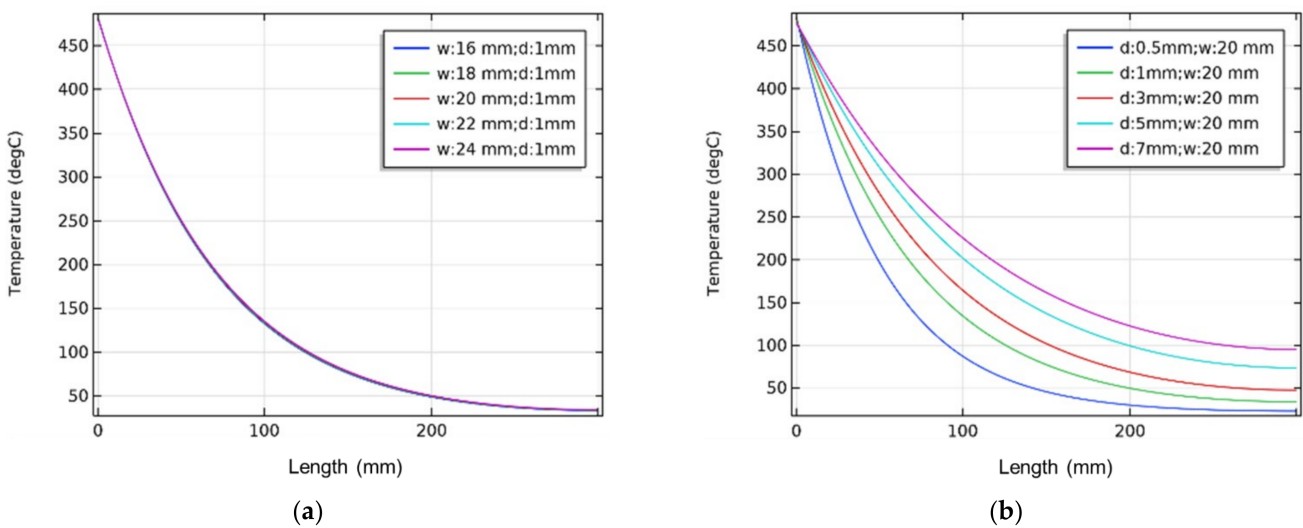

(**a**)                    (**b**)

**Figure 6.** Temperature buffer curves for waveguide rods of different sizes. (**a**) Temperature buffer curves for waveguide rods of different widths (16 mm, 18 mm, 20 mm, 22 mm, and 24 mm); (**b**) Temperature buffer curves for waveguide rods of different thickness (0.5 mm, 1 mm, 3 mm, 5 mm, and 7 mm).

In order to verify the simulation results, a test was conducted using an electric heating device, as shown in Figure 7. The thickness of the specimen was 50 mm, and the waveguide rod was 300 mm × 20 mm × 1 mm. The specimen was heated to be in a stable state of 500 deg C, and then an infrared temperature gun was applied to measure the temperature along the waveguide rod.

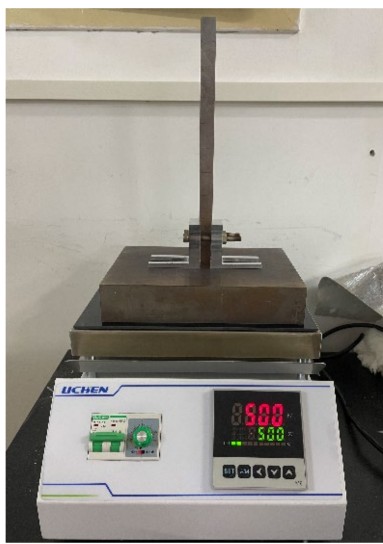

**Figure 7.** Electric heating device used in the test.

Temperature measurements were taken at 30 mm intervals on the waveguide rod using an infrared temperature gun. Figure 8 shows the comparison between the experimental results and the simulation results. Both the experimental and simulation results show the same trend of temperature decrease, and the temperature at the top of the waveguide rod is lower than 34 deg C. The above comparison shows the reliability of the thermal simulation.

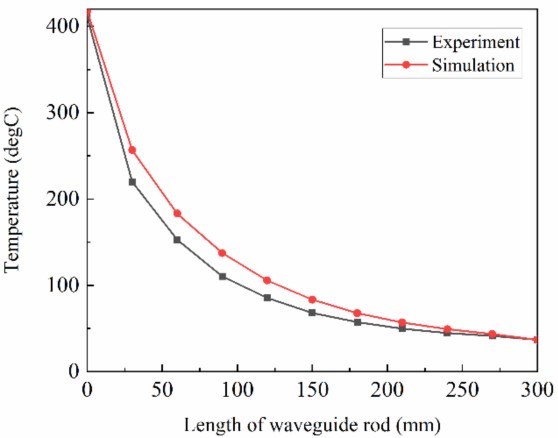

**Figure 8.** Comparison of calculated temperature buffer curve and measured temperature buffer curve.

## 3.2. Ultrasonic Fields

Ultrasonic propagation through the above-designed waveguide rod and specimen was simulated with the COMSOL Multiphysics. Figure 9 shows the local enlargement of the ultrasonic field at *y-z* plane, which gives the ultrasonic distribution in the waveguide rod and the specimen at a different time. It can be seen from the simulation diagram that the ultrasonic wave propagates along the waveguide rod to the inside of the test piece (Figure 9a), reflects and generates an echo at the bottom of the test piece (Figure 9b–d), and the reflected echo returns to the waveguide rod along the original path (Figure 9e,f). It also can be found that the ultrasonic waveguide undergoes a complex waveform conversion,

including longitudinal, transverse, and surface waves. Furthermore, the ultrasonic-guided wave undergoes significant attenuation and dissipation in the specimen, which is inevitable in the propagation process. However, the energy of the ultrasonic-guided wave is mainly concentrated in the center.

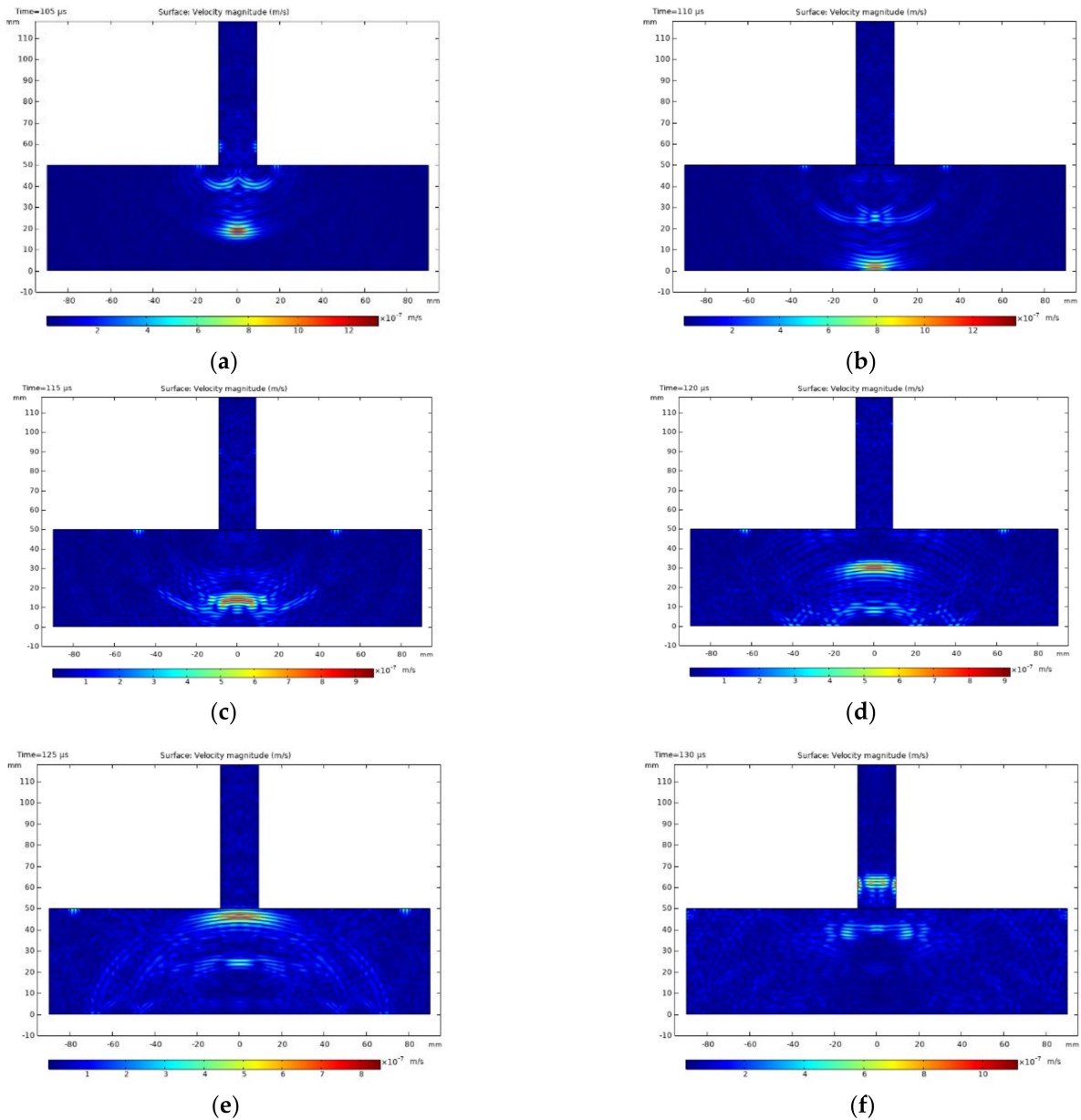

**Figure 9.** Ultrasonic field at different times. (**a**) $t$ = 105 μs; (**b**) $t$ = 110 μs; (**c**) $t$ = 115 μs; (**d**) $t$ = 120 μs; (**e**) $t$ = 125 μs; (**f**) $t$ = 130 μs.

Figure 10 shows the time-domain plot monitored by three probe points. The coordinates of the three probe points are (0, 0, 150), (0, 3, 150), and (0, 6, 150), which are all located on the horizontal centerline of the waveguide rod, but the distance from the vertical centerline of the waveguide rod is 0 mm, 3 mm, and 6 mm. As can be seen from the figure, the degree of signal attenuation and dispersion at different locations is different. The closer the receiving point to the center, the clearer the ultrasound signal. More serious ultrasound signal attenuation and dispersion occur as it moves gradually far away from the center. Therefore, probe 1 monitors the clearest ultrasound signal, and probe 3 monitors the most disturbed ultrasound signal. This suggests a conclusion that in order to obtain a clear

and discernible echo signal, the receiving point should be located on the centerline of the waveguide rod.

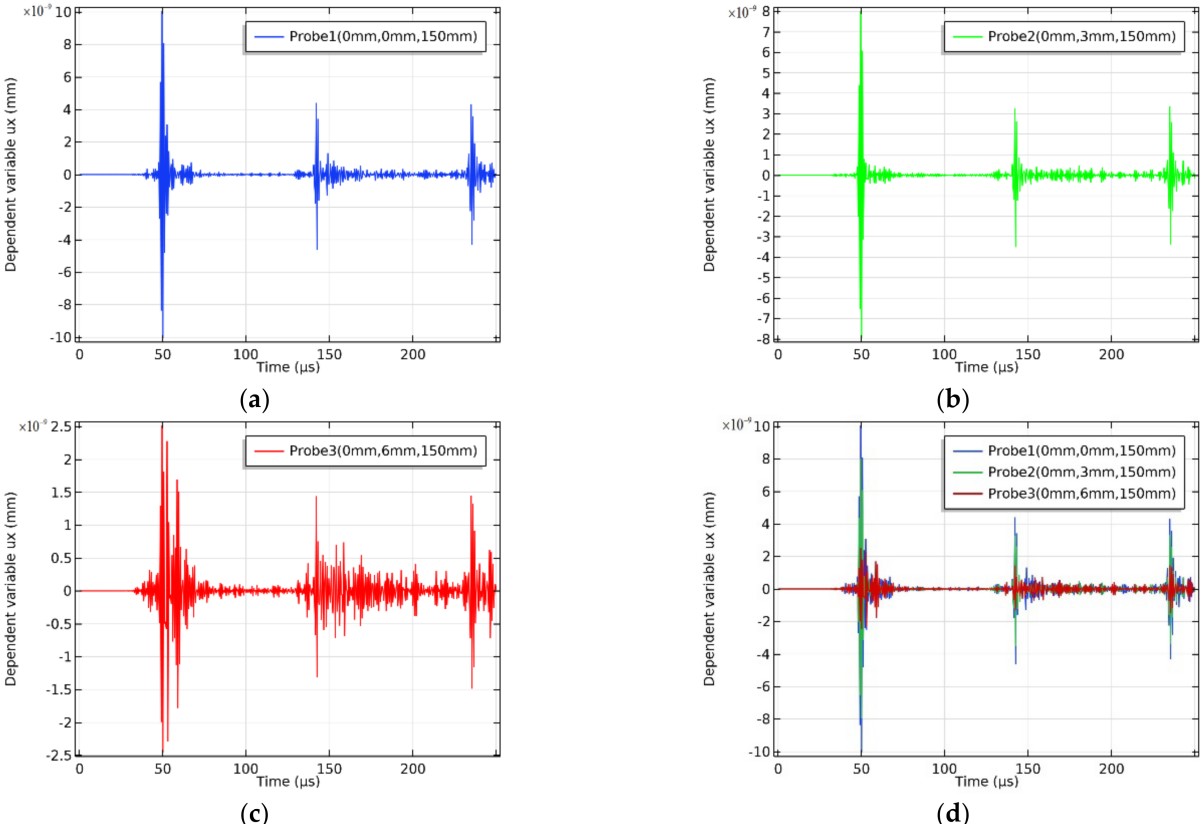

**Figure 10.** Time-domain plot of three probe points at the centerline of the waveguide rod. (**a**) Probe 1, coordinates (0 mm, 0 mm, 150 mm); (**b**) Probe 2, coordinates (0 mm, 3 mm, 150 mm); (**c**) Probe 3, coordinates (0 mm, 6 mm, 150 mm); (**d**) Probe 1 (**Blue**); Probe 2 (**Green**); Probe 3 (**Red**).

With the designed waveguide structure, an ultrasonic testing device was developed, as shown in Figure 11. It includes a signal generator, oscilloscope, waveguide structure, and clamping device. The end of the waveguide structure was fixed on the specimen by the clamping device with a strong pressing force. Therefore, the ultrasonic wave can transport between the specimen and the waveguide structure. The specimen was put on the electric heating device and heated to a certain temperature. Then, the device transmitted and received the ultrasonic signal. Two plate specimens were tested with a thickness of 20 mm and 50 mm, respectively. Figure 12 displays the received wave signals, and the crests of the echo are very clear. The thickness can be obtained according to the wave path difference. Figure 13 shows the measured data of the thickness at different temperatures. If the wave velocity is set as a constant, the predicted thickness basically increases linearly with the temperature. A large error occurs at high temperatures, which may reach to 6.9% and 5.0% at 250 deg C, respectively. Therefore, the effect of temperature must be considered in detecting the thickness of high-temperature objects.

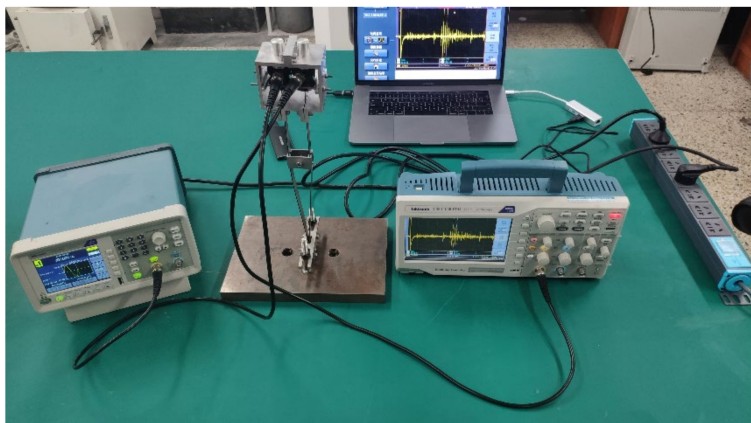

**Figure 11.** Self-developed ultrasonic-guided wave testing device.

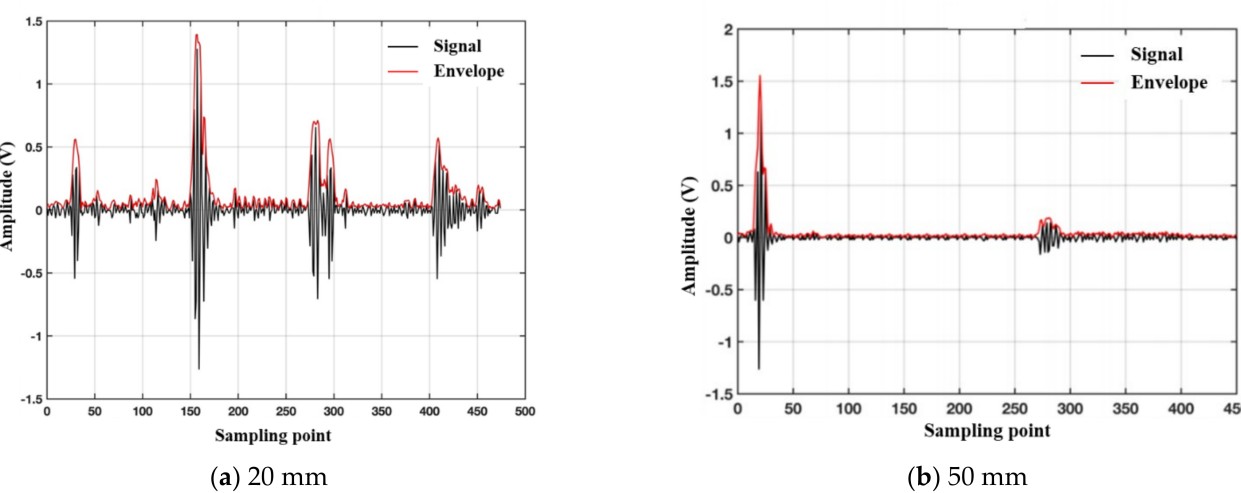

(**a**) 20 mm　　　　　　　　　　　　　　　　　　　(**b**) 50 mm

**Figure 12.** Received wave signals and envelope curves.

The specimen expands at high temperatures, and the thickness can be theoretically calculated based on the expansion coefficient. It is found that the theoretical thickness grows little below 250 deg C, which has a bigger difference with that measured at a constant wave velocity. To detect a more accurate thickness, a temperature-dependent method was applied, in which the ultrasonic wave velocity is not a constant and calculated by:

$$\mathbf{v} = \sqrt{G/\rho} \tag{11}$$

Here, $G$ is shear modulus and $\rho$ is density, and they both vary with temperature. Wave velocity was calculated by Equation (11), and it was used in the ultrasonic testing device to obtain the thickness at different temperatures. As shown in Figure 13, thickness predicted with the temperature-dependent wave velocity is close to the theoretical data. It indicates that the temperature-dependent method can achieve high precision detection at high temperatures.

The above research shows that the heat transfer model is useful to optimize the waveguide structure, and the designed ultrasonic-guided wave testing device is able to accurately measure the thickness of metals at high temperatures. Compared with the measurement at normal temperature, ultrasonic wave velocity varies with temperature so that the common measuring method with a constant wave velocity will cause a big error at high temperatures. The proposed temperature-dependent method, which considers the correlation of wave velocity and temperature, is necessary to achieve high precision detection at high temperatures. The research demonstrates its value in the thickness detection of high temperature objects.

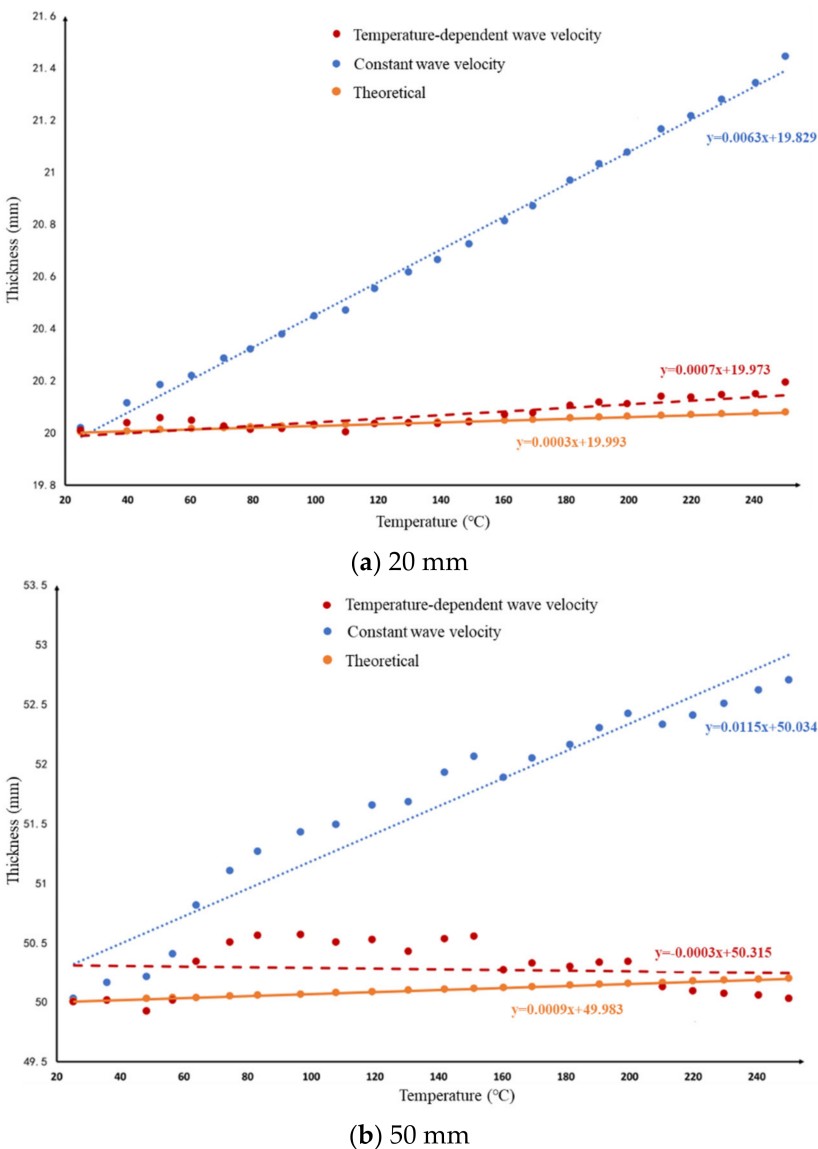

(**a**) 20 mm

(**b**) 50 mm

**Figure 13.** Predicted thickness at different temperatures.

### 4. Conclusions

A waveguide structure for temperature buffering was successfully designed to solve the problem that ordinary piezoelectric ultrasonic testing is not suitable for the high-temperature environment. Therefore, the paper designed a waveguide rod, determined its material, length and thickness with thermal simulations, and validated the feasibility with an experiment. It shows that SS316L is a good choice as the waveguide rod, and an optimum size is given to ensure the ultrasonic testing even at a high temperature of 500 deg C. The length is 300 mm, width 20 mm, and thickness 1 mm.

Ultrasonic propagation through the designed waveguide rod and specimen was calculated. Propagation, reflection, attenuation, and dissipation of the ultrasonic wave were revealed in the whole process. It shows that the closer to the center, the clearer the ultrasound signal, and serious ultrasound signal attenuation and dispersion occur as it gradually moves far away from the center. Therefore, the signal receiving point should be located at the centerline of the waveguide rod.

An ultrasonic-guided wave testing device was developed based on the designed waveguide structure. It measured the thickness of two specimens at different temperatures. It shows the detected thickness basically increases linearly with the temperature if the wave

velocity is regarded as a constant. It implies the measurement error also increases with temperature, which already reaches 6.9% and 5.0% at only 250 deg C for the two specimens. A temperature-dependent method was applied to achieve high precision detection at high temperatures, in which the ultrasonic wave velocity was related with temperature.

**Author Contributions:** Conceptualization, Methodology, Writing—review and editing, Project administration, Y.L.; Software, Writing—original draft preparation, Visualization, X.L.; Validation, R.L.; Data curation, Z.Y.; Investigation, Resources, J.Z. All authors have read and agreed to the published version of the manuscript.

**Funding:** This research was funded by the National Natural Science Foundation of China, grant number 52076216 and 51706246.

**Institutional Review Board Statement:** Not applicable.

**Data Availability Statement:** Data available on request from the authors.

**Conflicts of Interest:** The authors declare no conflict of interest.

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
