# Peer review of "Waveguide Structure Design and Simulation for High-Temperature Corrosion Thickness Detection"

_applsci, doi:10.3390/app122412960_

Round 1

Reviewer 1 Report

The authors reports that "waveguide structure for temperature buffering was successfully designed to solve the problem that ordinary piezoelectric ultrasonic testing is not suitable for the high-temperature environment". I cannot report the novelty in the setup, the presented setup is already used by other researchers including cited in this paper, see, e.g. review paper https://doi.org/10.3390/s21093200. 

Besides, the paper also have many other serious flaws, so it cannot be recommended for publication. Some major issues are given below:

- The temperature field is not taken into account in the model as it is described in the manuscript, whereas the presented structure is assumed to control thickness at higher temperatures.

- The experiment was also provided at regular temperatures, so it cannot validate the results for the stated problem.

- To simulate real situation, imperfect contact between sensor/waveguide and the inspected structures must be taken into account.

Author Response

  • The authors reports that "waveguide structure for temperature buffering was successfully designed to solve the problem that ordinary piezoelectric ultrasonic testing is not suitable for the high-temperature environment". I cannot report the novelty in the setup, the presented setup is already used by other researchers including cited in this paper, see, e.g. review paper https://doi.org/10.3390/s21093200.
  • Response:  

    It’s true that the presented setup is already used by other researchers. The goal of this research is to provide a simulation model to optimize the waveguide structure with the respect to its material, length, width and thickness. And we also supplemented more research about the high-temperature detection. An ultrasonic guided wave testing device was developed based on the designed waveguide structure. It measured the thickness of two specimens at different temperatures. It shows the detected thickness basically increases linearly with the temperature if the wave velocity is regarded as a constant. It implies the measurement error also increases with temperature, which already reaches to 6.9% and 5.0% at only 250 degC for the two specimens. A temperature-dependent method was applied to achieve high precision detection at high temperatures, in which the ultrasonic wave velocity was related with temperature.

  • Besides, the paper also have many other serious flaws, so it cannot be recommended for publication. Some major issues are given below:

    The temperature field is not taken into account in the model as it is described in the manuscript, whereas the presented structure is assumed to control thickness at higher temperatures.

  • Response: 

    We supplied more research about high-temperature detection. An ultrasonic guided wave testing device was developed based on the designed waveguide structure. We used it to detect the thickness at different temperatures.

  • The experiment was also provided at regular temperatures, so it cannot validate the results for the stated problem.

  • Response:

    We conducted high-temperature experiments in the revised manuscript. The experimental results validate our method.

  •  To simulate real situation, imperfect contact between sensor/waveguide and the inspected structures must be taken into account.
  • Response: In our experimental device, the waveguide structure was fixed on the specimen by a clamping device with a strong pressing force. So the contact is good for the two parts. In the simulation, the effect of the imperfect contact is not taken into account as it may have little impact. 

Reviewer 2 Report

This article has great scientific and research potential. However, there are parts that need to be corrected. First of all the introduction is weak. Most of the references are given in the range, for example “structural health monitoring 50 (SHM) [11-14]”, which suggests reference multiplication, which are not well known. It would be good to write at least a sentence or remark on each reference cited.

Computational method methodology is not well described. The scientific basis of methodology are described but without the references in relations to formulas used. I know that these are well-known formulas but still they need resources. When the device is shown it is not describe, the parameters of the device are not listed, the Figure itself do not give us an information. In Fig. 2 the “y” axis is not described.

Results are good, well described but there is no discussion. A discussion should be based on modern literature, especially when the results are based on numerical simulations is good to refer to similar results based on measurements. So you can refer to existing results and have a proper discussion, to see other researches results. It is also good to underline the novelty of these research. I do not see the novelty so far.

Author Response

  • This article has great scientific and research potential. However, there are parts that need to be corrected. First of all the introduction is weak. Most of the references are given in the range, for example “structural health monitoring 50 (SHM) [11-14]”, which suggests reference multiplication, which are not well known. It would be good to write at least a sentence or remark on each reference cited. 
  • Response: We revised the introduction part as you suggested. "UGWT can also be combined with the concept of structural health monitoring. Fromme et al. [11] developed a guided ultrasonic wave array to monitor structural integrity. Lugovtsova et al. [12] analyzed the guided wave propagation in a multi-layered structure. Diamanti et al. [13] applied UGWT to monitor the structural health of aircraft composite structures. Gao [14] conducted an in-depth study of ultrasonic guided wave mechanics in composite material. "
  • Computational method methodology is not well described. The scientific basis of methodology are described but without the references in relations to formulas used. I know that these are well-known formulas but still they need resources. When the device is shown it is not describe, the parameters of the device are not listed, the Figure itself do not give us an information. In Fig. 2 the “y” axis is not described. 
  • Response: 

    We cited the references [23,24] in relations to formulas used.

    The parameters of the device are not directly given because the goal of this research is to optimize the size of the device. So we used various parameters to test in the simulation, and finally determined the waveguide rod parameters as 300 mm X 20 mm X 1 mm.

    We revised the Fig.2 as you suggested.

  • Results are good, well described but there is no discussion. A discussion should be based on modern literature, especially when the results are based on numerical simulations is good to refer to similar results based on measurements. So you can refer to existing results and have a proper discussion, to see other researches results. It is also good to underline the novelty of these research. I do not see the novelty so far.
  • Response: 

    We supplemented more research about the high-temperature detection. An ultrasonic guided wave testing device was developed based on the designed waveguide structure. It measured the thickness of two specimens at different temperatures. Some new findings are supplied, and more discussion is made. It shows the detected thickness basically increases linearly with the temperature if the wave velocity is regarded as a constant. It implies the measurement error also increases with temperature, which already reaches to 6.9% and 5.0% at only 250 degC for the two specimens. A temperature-dependent method was applied to achieve high precision detection at high temperatures, in which the ultrasonic wave velocity was related with temperature.

Reviewer 3 Report

The authors presented a study on the Waveguide Structure Design and Simulation for High-Temperature Corrosion Thickness Detection.

The following comments should be addressed before the acceptance of the paper:

-        The introduction is relatively short and should be extended.

-        What is the novelty of the work? To be clearly stated.

-        The governing equations and numerical method are to be detailed.

-        What is the convergence criterion ?

-        The used assumptions are to be detailed.

-        What is the considered volumetric force?

-        The boundary conditions are to be expressed mathematically.

-        The authors presented results related to the temperature without presenting the energy equation.

-        A figure presenting the used mesh is to be added.

-        A grid sensitivity test is to be performed.

Author Response

The authors presented a study on the Waveguide Structure Design and Simulation for High-Temperature Corrosion Thickness Detection.

The following comments should be addressed before the acceptance of the paper:

  • The introduction is relatively short and should be extended.
  • Response:  We extended the introduction and supplied more content about experiment.
  • What is the novelty of the work? To be clearly stated.
  • Response:  

    The paper applies numerical simulation to design the waveguide structure for ultrasonic testing at high temperatures. The model can optimize the waveguide structure with the respect to its material, length, width and thickness. An ultrasonic guided wave testing device was developed based on the designed waveguide structure. It measured the specimen thickness at different temperatures, and some new findings are revealed. It shows the detected thickness basically increases linearly with the temperature if the wave velocity is regarded as a constant. It implies the measurement error also increases with temperature. A temperature-dependent method was applied to achieve high precision detection at high temperatures, in which the ultrasonic wave velocity was related with temperature.

  • The governing equations and numerical method are to be detailed.
  • Response: We supplied the energy equation and mesh model.
  • What is the convergence criterion ?
  • Response:  Default criterions in the software are applied in the calculation. It’s 10-6 for the energy equation, and 10-3 for the others.
  • The used assumptions are to be detailed.
  • Response:  We supplied the assumption about heat transfer at the surfaces. Also, there are new contents about contact surface between waveguide structure and specimen.
  • What is the considered volumetric force?
  • Response:  There is no volumetric force in the model.
  • The boundary conditions are to be expressed mathematically.
  • Response:  We expressed the thermal boundary conditions mathematically in equation (3).
  •  The authors presented results related to the temperature without presenting the energy equation.
  • Response:  The energy equation is supplied in the revised manuscript.
  •  A figure presenting the used mesh is to be added.
  • Response: The mesh figure is supplied in the revised manuscript.
  • A grid sensitivity test is to be performed.
  • Response:  Grid sensitivity tests were made, and the calculated result is correct.

Round 2

Reviewer 1 Report

The authors have explained the details of the model and sufficiently improved the whole manuscript.

Author Response

Thank you very much for your review and kind suggestions.

Reviewer 2 Report

It is much better version of the article, but the discussion is still missing, but now I think it could be published

Author Response

Thank you very much for your review and kind suggestions. We added a new paragraph in the manuscript to discuss our key findings.

"Above research shows that heat transfer model is useful to optimize the waveguide structure, and the designed ultrasonic guided wave testing device is able to accurately measure the thickness of metals at high temperatures. Compared with the measurement at normal temperature, ultrasonic wave velocity varies with temperature so that the common measuring method with a constant wave velocity will cause a big error at high temperatures. The proposed temperature-dependent method, which considers the correlation of wave velocity and temperature, is necessary to achieve high precision detection at high temperatures. The research demonstrates its value in the thickness detection of high temperature objects."

Reviewer 3 Report

After revision, the paper can be accepted for publication

Author Response

(The authors gave the same response as above.)
